

# Framework to perform taint analysis and security assessment of IoT devices in smart cities

Akashdeep Bhardwaj[1], Ankit Vishnoi[2], Salil Bharany[3], Abdelzahir Abdelmaboud[4], Ashraf Osman Ibrahim[5], Mohamed Mamoun[6] and Wamda Nagmeldin[7]

[1] School of Computer Science, University of Petroleum and Energy Studies, Dehradun, India
[2] Computer Science and Engineering Department, Symbiosis Institute of Technology (SIT), Symbiosis International (Deemed) University (SIU), Pune, India
[3] Department of Computer Science and Engineering, Lovely Professional University, Phagwara, Punjab, India
[4] Department of Information Systems, King Khalid University, Muhayel Aseer, Saudi Arabia
[5] Universiti Malaysia Sabah, Sabah, Malaysia
[6] Alzaiem Alazhari University, Khartoum North, Sudan
[7] Prince Sattam bin Abdulaziz University, Al-Kharj, Saudi Arabia

Corresponding authors
Salil Bharany,
salil.bharany@gmail.com
Ashraf Osman Ibrahim,
ashrafosman@ums.edu.my

## ABSTRACT

The Internet of Things has a bootloader and applications responsible for initializing the device's hardware and loading the operating system or firmware. Ensuring the security of the bootloader is crucial to protect against malicious firmware or software being loaded onto the device. One way to increase the security of the bootloader is to use digital signature verification to ensure that only authorized firmware can be loaded onto the device. Additionally, implementing secure boot processes, such as a chain of trust, can prevent unauthorized access to the device's firmware and protect against tampering during the boot process. This research is based on the firmware bootloader and application dataflow taint analysis and security assessment of IoT devices as the most critical step in ensuring the security and integrity of these devices. This process helps identify vulnerabilities and potential attack vectors that attackers could exploit and provides a foundation for developing effective remediation strategies.

## INTRODUCTION

The market for IoT devices has expanded rapidly in recent years. To be competitive, time-to-market has become critical; the sooner a rival makes and combines his/her product, and more inclined he/she is to lead the market. Due to a lack of validation or quick turnaround time, this rivalry causes severe software flaws in the systems. Many expose flaws that could be exploited by botnet or malware attacks. Furthermore, they are vulnerable to many zero-day attacks that need immediate intervention to preserve the privacy of the system where the IoT device is placed. The most effective way to fight these attacks is to quickly upgrade the software of such devices *via* patches. A crucial component known as the bootloader

must be installed throughout this procedure to run the embedded system's setup, control, and supervision. This code can control and perform the boot sequence as well as run the firmware. However, in the absence of any guidelines or references, there is presently no generic bootloader for all IoT devices, rather there are various bootloaders specialized to a specific set of hardware or kernel.

IoT devices rely on firmware to function properly and securely. The firmware bootloader, which is responsible for initializing and managing the device's hardware and firmware, plays a critical role in ensuring the security and integrity of IoT devices. The unique firmware bootloader analysis and security assessment of IoT (Internet of Things) devices is essential to identify vulnerabilities and potential attack vectors that could be exploited by attackers. A bootloader (*IoT ONE*) is a program that runs on an IoT device before the main operating system is loaded. It is responsible for initializing the hardware and loading the operating system into memory. Bootloaders play a critical role in the security and functionality of IoT devices (*Gillis, 2022*) as they ensure that only authorized software is loaded and executed on the device. In embedded and IoT devices, a bootloader is a small program that is stored in a non-volatile memory, such as random memory or flash memory. Its main function is to initialize the device's hardware and load the main operating system or firmware into memory. The bootloader is executed immediately after the device is powered on or reset, and it runs before the main operating system or firmware.

The firmware is a low-level program that controls access to an IoT device's hardware and peripherals as well as offering a variety of services to higher-level apps. There are three components to firmware:

- **Bootloader** (*IoT ONE*) is a low-level software that loads the primary operating system and initializes the hardware. It is the first program run when a device is turned on or after a reset. It runs in two stages, with the first loading basic code and the second loading the IoT OS. By doing this, the second stage gets updated while the first stage is kept unchanged.
- **Operating system** (*Arm Ltd*) offers an environment in which applications can run. The bootloader loads and launches the OS kernel, which is the fundamental part of the operating system. Operating systems might have security flaws much like the bootloader, but locating these is likewise not simple.
- **Device file system** (*Gillis, 2022*) is where configuration settings, libraries, development environments, and programs are kept which are pre-installed with web servers, enabling web-based remote configuration of the device. Such applications are of particular interest to hackers because it is not necessary to have specialized knowledge of embedded systems to uncover flaws in them.

Research gaps in this area include the need for more secure and efficient bootloading mechanisms for IoT devices, as well as the need for better tools and techniques for analyzing and understanding the inner workings of proprietary bootloaders. Additionally,

there is a need for more research on the impact of different types of attacks on the bootloading process, and the development of countermeasures to protect against these attacks. Reviewing the gaps, the highlights of this research are to enhance the state of firmware security by discovering new security vulnerabilities using unique tools and by decreasing the threat surface area and presenting new tools to discover bugs in embedded device bootloader, perform code de-bloating on firmware binaries and fuzz IoT devices.

This study tries to answer some questions helping in determining the design and methods to guide the search phases in this study as:

- Current vulnerabilities and potential threats related to the current IoT device bootloading, mainly in the context of firmware and application dataflow taint analysis.
- Limitations or potential exploitations of digital signature verification approaches, and how effectively they can validate and integrity of installed firmware and software on IoT devices.
- Difficulties and knowledge gaps exist regarding the proprietary bootloaders utilized in Internet of Things devices and the impact of the process on overall security.
- How IoT devices boot up is impacted by different attack kinds; learn which are the strongest countermeasures against these potentially dangerous attacks.

To satisfy the research gaps, this research attempts to present an overview of the distinct firmware bootloader taint analysis and security evaluation procedure for IoT devices. An in-house Python taint analysis tool is used to thoroughly examine the bootloader code and its interactions with the device's firmware and hardware at the start of the procedure. Analyzing the peripheral initialization routines, communication interfaces, and memory management of the bootloader are all included in this. The goal of the analysis is to find any potential weaknesses that an attacker could exploit, like memory leaks, buffer overflows, or uninitialized variables. Furthermore, since the bootloader is a popular attack vector for Internet of Things devices, its handling of firmware updates is also assessed. The next stage is to evaluate the bootloader's security, which entails assessing its resistance to popular attack techniques such as denial-of-service attacks, code injection, and firmware modification. Following the analysis and security assessment, remediation methods are created to fix any vulnerabilities found and raise the device's overall security.

## RELATED WORK

A bijective time-stamped technique for identifying IoT device software was introduced by *Urien (2020)*, with a focus on memory space and constant computing time. The approach uses a hash function and a normal distribution to compute a memory fingerprint. To meet the demand for secure firmware upgrades in low-cost embedded solutions, *Jaouhari & Bouvet (2022)* developed a generic bootloader for OTA updates in IoT devices based on FreeRTOS. Attackers can take advantage of security flaws in IoT device firmware upgrades and bootloaders, as presented by *Morel & Couroussé (2019)*. *Romana, Grandhi & Eswari (2020)* performed a security analysis on a particular router and provided a technique for assessing the security aspects of SOHO routers. *Anand & Premananda (2022)* proposed a

**Table 1 Summary of references.**

| Reference | Pros | Cons |
|---|---|---|
| *Urien (2020)* | • Detects corrupted software in IoT devices to ensure software integrity and security.<br>• The algorithm relies on two aspects–the memory space is finite, and the computing time is stable, which could make it more reliable than other methods.<br>• The algorithm computes a memory fingerprint with a hash function, according to a pseudo-random order, fixed by a permutation P, which could make it more difficult for attackers to bypass.<br>• The source code is open and published, which could make it easier for other researchers to build upon and improve. | • The implementation demonstrated on Arduino Nano 3.x powered by the ATmega328 processor limits the applicability to other IoT devices.<br>• The algorithm assumes that the decompression operations imply delays, which may not always be the case.<br>• The algorithm's computing time follows a normal distribution, which means that it may not be as effective at detecting certain types of attacks that do not significantly impact computing time.<br>• The algorithm's use of permutations may make it more complex and difficult to implement than other methods. |
| *Jaouhari & Bouvet (2022)* | • State of art and a comparison of some popular bootloaders currently used in constrained IoT devices is presented here.<br>• Generic bootloading process for typical IoT devices is discussed. Proof of Concept of the firmware over the air process, which uses the generic bootloader on top of ones of the most used OS (*i.e.*, FreeRTOS).<br>• The discussion of a secure and generic bootloading process that guarantees the integrity and the authenticity of the received firmware image. | • This research mentions there are several drawbacks in the current proposition that require deeper investigations to provide a generic, portable, robust, and secure bootloader for IoT and for the constrained ones.<br>• This research discussed only few drawbacks and, most of them were the ones related to security and evaluations. |
| *Morel & Couroussé (2019)* | • Proposed security mitigation for hardware attacks like prevention of fault injection attacks using secure bootloaders that verify the integrity of the firmware before execution is presented here.<br>• Physical attacks such as side-channel attack countermeasures were proposed such as masking, shuffling, or randomizing the data.<br>• For mitigating software attacks, control-flow integrity, control-flow attestation, stack canaries, and address-space layout randomization were the proposed countermeasures that can be used to prevent or mitigate attacks.<br>• Cryptographic primitives such as cipher keys and authentication codes can be protected against timing attacks by ensuring that the executed code is not dependent on the data being processed. | • Physical attacks, such as fault injection attacks can easily bypass security mechanisms and gain access to sensitive data.<br>• Software attacks, such as buffer overflow attacks can be used to execute malicious code and gain control of the device.<br>• This research does not discuss how attackers attempt and intercept firmware updates in transit, modify the firmware, and then install the modified firmware on the device. |
| *Romana, Grandhi & Eswari (2020)* | • Safeguarding of SOHO devices by re-configuring them for reasonable security is proposed in this research. This is especially important because many users deploy these devices with insecure default configurations, leaving them vulnerable to attacks.<br>• Enabling advanced threat mitigation techniques for these devices, which are otherwise available to personal computers, is a challenge because of the limited processing, memory, storage. Therefore, users should take the time to learn about the security features of their SOHO routers and configure them appropriately to ensure that they are protected from potential threats. | • Devices can become easy targets for attackers due to their easy exploitability, making them an attackers' paradise.<br>• There have been numerous reports of security issues in SOHO routers because of known vulnerabilities, which can lead to unauthorized access, data theft, and other malicious activities.<br>• The vendors often overlook the security of these devices and sell them with default insecure settings, which can leave users vulnerable to attacks.<br>• Even if vendors release security patches, very few devices end up getting installed with these patches, which can leave users exposed to known vulnerabilities |

| Reference | Pros | Cons |
|---|---|---|
| *Anand & Premananda (2022)* | • The advantage of establishing security between servers, cloud applications, and users in today's world is well reviewed in this research, as the number of devices connecting to the internet is ever increasing.<br>• This means that there is a greater risk of cyberattacks and data breaches, which can have serious consequences for individuals and organizations.<br>• The article establishes critical security aspects to protect sensitive information and ensure the safe and reliable functioning of internet-connected devices and services. | • The proposed model uses external flash to boot user applications which slows down the processing speed of the execution of the application and boot loading time.<br>• This is because the external flash is used to boot the user applications, which takes more time compared to booting from RAM. |
| *Zhu et al. (2020)* | • The research approach in this article uses to firmware code analysis which differs from traditional methods by breaking away from the traditional feature-centered approach and focusing on code classification and the qualitative description of code features to discuss the idea of code similarity and homology analysis.<br>• The proposed approach is information-centric, focusing on the informativeness (essentiality, stability, anti-variability, and heritability) of the firmware code genes and the quantitative analysis of firmware code similarity and homology by discussing common methods and mechanisms. | • Two major challenges associated with detecting security risks in IoT firmware.<br>• First is heterogeneity and closed source, where the firmware of an IoT terminal is deployed in various architectures, with different instruction sets, registers, addressing modes, stack management, calling conventions, storage management models. Most firmware has closed-source code, is unable to obtain the source code, and lacks symbol debugging information. Thus, security detection objects of terminal firmware are not unified, and detection is difficult.<br>• Second is limited resources, where most IoT terminals belong to the category of embedded devices, with limited storage and computing resources, and many terminals have high requirements for power consumption and real-time performance. Therefore, it is difficult for the terminal itself to deploy antiviral, intrusion detection and other security protection measures. Additionally, it is difficult to adopt underlying monitoring, probing of early warnings and other security monitoring means. |
| *Zhu et al. (2019)* | • The unique approach presented in this article differs from other firmware security detection technologies based on similarity by attempting to address this issue theoretically.<br>• This new approach detected security risks in IoT terminal firmware by mining firmware code genes, which can essentially identify code and exhibit stability, anti-variability, and heritability. This approach provides a foundation for cross-platform firmware binary code homology and similarity analysis. | • Use of COTS proposed in this research has high code reuse rates. Such firmware is always heterogeneous and closed source, this makes it difficult to detect and investigate the security risks at the firmware level that their impacts are faster and broader.<br>• High code reuse rates in IoT terminal firmware make it difficult to detect and investigate security risks at the firmware level, which can have significant impacts on the security of connected devices and networks. |
| *Choi & Lee (2020)* | • This blockchain-based distributed firmware update architecture offers several advantages compared to the traditional client-server model.<br>• Proposed architecture provides decentralization, transparency, and irreversibility, which are characteristics of blockchain technology. | • Firmware update of an IoT device is necessary for its lifecycle, and secure firmware update of the IoT device is being brought as the first step in IoT security.<br>• Firmware update failures can occur due to network issues or cyber-attacks, support for integrity and authentication of the firmware images are required. |

*(Continued)*

| Reference | Pros | Cons |
|---|---|---|
| | • This blockchain-based distributed firmware update architecture offers several advantages compared to the traditional client-server model.<br>• Proposed architecture provides decentralization, transparency, and irreversibility, which are characteristics of blockchain technology.<br>• The distributed nature of the architecture ensures that every node stores the same data based on an append-only distributed ledger, which provides integrity, decentralization, and irreversibility.<br>• This approach can prevent targeting issues and author-disappearing issues, which are not addressed by the current SUIT working group's traditional client-server model.<br>• The proposed architecture is also tolerant to a single point of failure and enables irreversible downloads even in the author-disappearing state. | • Firmware update of an IoT device is necessary for its lifecycle, and secure firmware update of the IoT device is being brought as the first step in IoT security.<br>• Firmware update failures can occur due to network issues or cyber-attacks, support for integrity and authentication of the firmware images are required.<br>• Firmware updates for IoT devices are vulnerable against an author-disappearing issue that the IoT device manufacturers or firmware vendors are unable to provide firmware updates in time due to cyber-attacks or disappearing due to their funding problems. |
| *Zandberg et al. (2019)* | • The article presented several experimental results to measure and compare the performance of various crypto libraries that are relevant in the context of secure firmware updates for constrained IoT devices.<br>• The performance of several deployment configurations using their prototype and provide the first experimental evaluation of the IETF SUIT specification is presented.<br>• The results displayed that the prototype could provide secure firmware updates on a large variety of constrained IoT devices, while entirely avoiding proprietary mechanisms and code. | • IoT devices without a built-in firmware update mechanism are vulnerable to security threats such as large-scale DDoS attacks using compromised IoT devices,<br>• Software-based attacks such as buffer overflow attacks are on the rise and work on memory isolation or compartmentalization is pending.<br>• Firmware updates can themselves become attack vectors if not designed correctly, as demonstrated by the Zigbee Worm. |
| *Wang et al. (2019)* | • The advantages of firmware vulnerability detection are achieving large-scale firmware security inspection accurately and efficiently.<br>• The proposed method detected vulnerabilities in firmware images without access to the source code, and it can identify vulnerabilities that are caused by code reuse.<br>• The method also detected vulnerabilities that are not detected by traditional methods, such as signature-based methods and anomaly-based methods. | • The endless emergence and ubiquitous deployment of IoT devices have exposed a significant number of potential targets to the outside world.<br>• IoT devices have become one of the most popular targets for hackers and one of the easiest to attack, as proven by the increasing attacking events targeting IoT devices in recent years.<br>• IoT vendors tend to reuse easy-to-obtain yet unsafe software modules in their device firmware, and vulnerabilities in certain software modules may affect large number of IoT devices. |
| *Kim et al. (2021)* | • FIRM-COV was able to find the fastest and most 1-day vulnerabilities with almost no false-positives.<br>• This research also found two 0-day vulnerabilities in real-world IoT devices within 24 hours. | • FIRM-COV proposed an optimized emulation of IoT firmware to detect vulnerabilities without requiring real-world devices by applying two emulations.<br>• It generally executes the target program in user-mode emulation for efficiency, however exceptions are caused in the system if t switched to full-system emulation to handle exceptions.<br>• Only after optimizing the existing emulation technique, FIRM-COV could maintain a stable state and achieves high accuracy when detecting vulnerabilities. |

| Reference | Pros | Cons |
|---|---|---|
| *Gui et al. (2020)* | • FIRMCORN optimized the initial environment of virtual execution by using the real IoT device dump context.<br>• This used heuristic algorithms to search the three types of functions to optimize the virtual execution process, thereby achieving faster, more accurate, and more stable virtual execution. | • FIRMCORN focused only on three IoT firmware fuzzing issues namely high throughput required by fuzzing, inaccuracy of emulation compared with real devices, and instability of emulation due to lack of hardware.<br>• This research had a limited scope. |
| *Yu et al. (2020)* | • The proposed approach to firmware identification works by adopting the widespread weak password technology available for online IoT devices to grab web content, which does not interfere with the normal operation of the IoT device.<br>• The method consisted of three steps: coarse-grained identification to identify the brand of the device, identifying the location of the navigation bar of the firmware version to accurately obtain the firmware version, and dividing the webpage into blocks, filtering out redundant pages, obtaining the main page where the firmware version is located, and extracting the device model and firmware version through regular expressions.<br>• The experimental results show that the method achieves 95.97% accuracy in device firmware identification, superior to other methods. | • This research only discussed risk of information security due to growth of Internet-connected IoT devices.<br>• The authors only provided examples of large-scale network disconnection caused by the Mirai infection in 2016, the iot_reaper attack in 2017, and the VPNFilter malware in 2018 but not the actual methodology or ways to replicated.<br>• The vulnerabilities exploited by attackers are closely related to the device firmware version. This research did not identify devices firmware version as an essential prerequisite for maintaining device security. |

solution to improve the bootloading time in IoT devices, enhancing their performance. *Zhu et al. (2020)* introduced an information-centric approach for analyzing firmware code similarities and homology. *Zhu et al. (2019)* explored firmware code genes for identifying code and assessing their stability, anti-variability, and heredity. *Choi & Lee (2020)* suggested a distributed patch management architecture using blockchain to enhance firmware upgrade security. *Zandberg et al. (2019)* reviewed guidelines and libraries for secure firmware upgrades in limited-power IoT devices. *Wang et al. (2019)* proposed a staged firmware vulnerability detection method based on code similarity. *Kim et al. (2021)* introduced a high-surveillance grey box fuzzer for IoT firmware to identify real-world vulnerabilities. *Gui et al. (2020)* developed a fuzzer tailored for IoT firmware vulnerability identification, addressing key challenges. *Yu et al. (2020)* presented a method for determining IoT device software using website page data and weak passwords. *Ebbers (2022)* analyzed firmware upgrades on IoT devices using data mining and mapping techniques. *Feng et al. (2023)* examined challenges and solutions for firmware security analysis in IoT devices. *Hassija et al. (2019)* discussed security challenges and emerging technologies to enhance trust levels in IoT applications. *Ammar, Russello & Crispo (2018)* surveyed the security aspects of prominent IoT frameworks, emphasizing architectural designs and security features. *Nebbione & Calzarossa (2020)* examined security within application layer protocols, addressing key challenges and best practices. *Khan, Awang &*

*Karim (2022)* conducted a comprehensive review of IoT security, focusing on wireless communication methods and technologies. *Roopak, Yun Tian & Chambers (2019)* introduced deep learning models for IoT cybersecurity, outperforming traditional machine learning algorithms. *Sicari et al. (2022)* explored the Function as a Service paradigm for creating scientific workflows. *Celesti et al. (2020)* proposed a telemedical laboratory service using IoT devices and Cloud computing for healthcare collaboration among professionals.

The research on IoT security encompasses vulnerabilities, mitigation strategies, and innovations, primarily in bootloader security, firmware updates, and network complexities. Challenges include resource limitations, closed-source firmware, and balancing security with device performance. Emerging trends emphasize machine learning and firmware security analysis for more robust security measures.

Table 1 summarizes the pros and cons of the top 12 research manuscripts relevant to this research.

## MATERIALS AND METHODOLOGY

Tools for IoT bootloader analysis are frequently used to examine IoT device bootloaders. These tools are used to look through the device's firmware and see if there are any security flaws or vulnerabilities that need to be fixed. They can also be used to retrieve device-specific data, like hardware specs, manufacturer, and version. Firmware analysis tools like Firmadyne, firmware reverse engineering tools like IDA Pro, and firmware security assessment tools like Binwalk are a few examples of IoT Bootloader Analyzer tools. To guarantee the security and integrity of IoT devices, security researchers, IoT device manufacturers, and other experts employ these techniques. Tools for "taint analysis" are used to examine the security of bootloaders on Internet of Things devices. They can be used to extract and examine firmware images, find security flaws, and check if standard security features like firmware signing and secure boot are present. The security of IoT device bootloaders is examined by this utility. Firmware images may be extracted and analyzed, vulnerabilities can be found, and standard security features like secure boot and firmware signing can be tested for. With the aid of these instruments, firmware reverse engineering and firmware image modification with the addition of unique payloads or patches are possible.

Depending on the firmware image being examined, different undiscovered vulnerabilities may be found, however in general, a variety of problems might be found, such as:

- Buffer overflow vulnerabilities: These arise when an application attempts to store more data in a buffer than its capacity permits, resulting in the data spilling over into neighboring memory regions.
- Hardcoded credentials refer to passwords or keys that are inherently incorporated in firmware, making them easily retrievable by an adversary.
- Insecure communication: This can involve using keys or passwords that are simple to figure out or unencrypted communication protocols.

- Inadequately secured storage: This can involve keeping private information in plaintext files or other easily accessible, unencrypted places.
- Insecure updates: Using unencrypted or unauthenticated update protocols puts the device at risk of malware being installed by an attacker.
- Privilege escalation is the process by which an attacker uses a firmware flaw to obtain access to higher-level privileges than they should.
- Unauthorized access: This can include the use of easily guessable default credentials or the lack of proper access controls in the firmware.
- Weak encryption: This can include the use of easily crackable encryption algorithms or the use of easily guessable encryption keys.

These are just some examples of the types of vulnerabilities that the tool may detect, and the actual vulnerabilities that are found will depend on the specific firmware image being analyzed. Overall, such tools are hugely valuable for security researchers and IoT device manufacturers looking to secure the boot process of their devices with the proposed algorithm including the below-mentioned steps:

i) Extracting the firmware image from the IoT device.
ii) Identifying the type of processor and operating system used in the device.
iii) Analyzing the firmware for known vulnerabilities and common security features such as secure boot and firmware signing.
iv) Decompiling the firmware to extract the underlying source code.
v) Performing firmware reverse engineering to identify additional vulnerabilities.
vi) Adding patches or unique payloads to the firmware.
vii) Checking for security flaws in the bootloader and firmware.
viii) Writing a report outlining the analysis's conclusions and emphasizing any security concerns or vulnerabilities found.

The setup consists of layers involving embedded sensors and actuator devices connecting to the physical world providing the status of the physical state changes. These devices are locally connected through a gateway that in turn connects to the Internet or the IoT Cloud platform. The cloud platform runs applications that remotely want to supervise and manage the physical IoT devices. Figure 1 illustrates this setup further.

This setup translates the data into different layers, the physical IoT device runs the embedded software along with edge software on the local gateway as illustrated in Fig. 2. The IoT backend communicates with the cloud storage services and databases that in turn access the Internet *via* web apps instead of web services for other purposes like Java Servlet, JSP Pages, or Android apps. In IoT, different nodes of the scheme have different software even as very few layers inside IoT devices communicate with the external world in terms of physical access.

The research methodology followed for analyzing taint analysis and IoT firmware typically involves several stages:

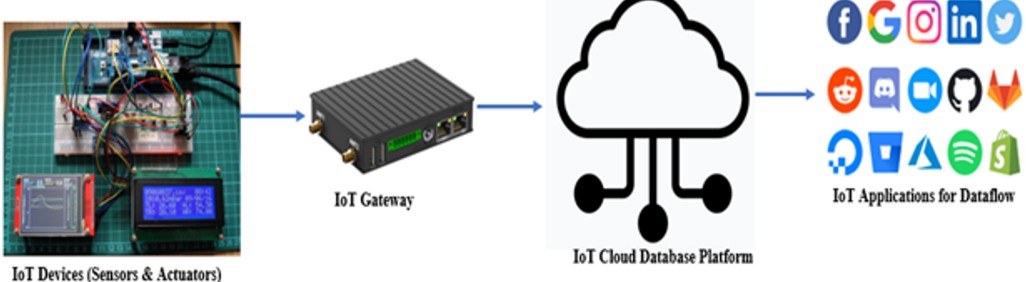

**Figure 1 IoT, cloud, and application connectivity.**

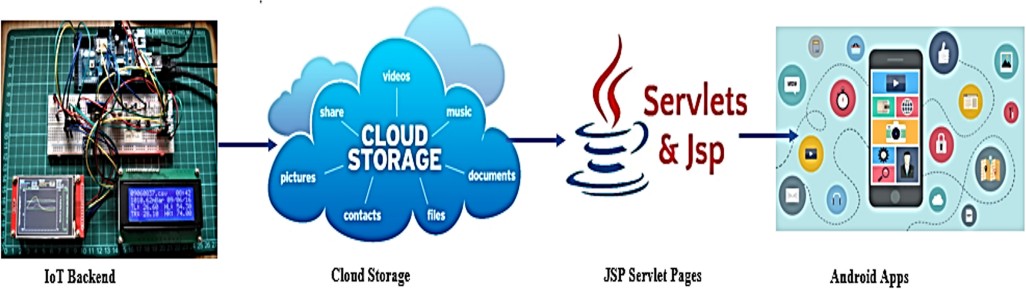

**Figure 2 IoT physical to logical dataflow.**

i)  Taint analysis discovers some anomaly from a source flow to a sink. IoT systems have at least two different types of sources and sinks.

   a) The first is the external components interacting with the physical real world where sensors as the source and actuators are the sink or the Android apps for geolocation as the source and set the label as the sink.

   b) The second uses a database containing the outcome as well as communication routes as the sink or the Internet with the request and receive as the source. obtain as the answer and source. as the sink.

ii)  Firmware extraction involves removing firmware images from a variety of IoT devices, including those with embedded Linux, RTOS, and microcontroller systems. After extracting the firmware image, analysis and examination are performed to find security flaws and vulnerabilities. This includes the device's operating system and processor type as well as any open network ports, hard-coded emails/passwords, and other security issues.

iii)  Firmware reverse engineering extracts the underlying source code and decompiles the firmware, undertaking firmware reverse engineering. This makes it possible for security experts to do a more in-depth analysis of the firmware and find any potential flaws.

iv)  Modification can be done by adding patches or unique payloads to firmware images. This can be used to add unique features to the device or test the security of the device.

v) By conducting attacks on the firmware and bootloader, test and evaluate the security of these systems.

vi) Provide reports detailing the firmware's testing and analysis, emphasizing any security concerns or vulnerabilities that were found.

The steps for the Taint Analysis algorithm proposed in this research for analyzing IoT firmware are explained in the pseudocode below:

i) The firmware image is extracted from the IoT device using the function 'extract-firmware' in the first line of the pseudocode.

ii) The firmware image is examined using the 'analyze-firmware' function in the following line. The CPU and operating system type of the device would probably be determined by this function, which would also look for known vulnerabilities and standard security measures like firmware signing and secure boot.

iii) Firmware reverse engineering is performed on the firmware image using a function named 'reverse-engineer-firmware'. To retrieve the firmware's underlying source code and find more vulnerabilities, this function decompiles the firmware.

iv) Next a function called 'patch-firmware modifies the firmware image to include custom payloads or patches and the 'test-firmware' function tests the firmware and bootloader for security vulnerabilities.

v) Finally, the function 'generate-report is called for generating a report on the findings of the analysis, highlighting any vulnerabilities or security risks that were identified.

The implementation of the algorithm is complex since the lines of code, functions, and parameters depend upon the actual implementation, which the authors have witnessed to vary depending on the specific implementation of the proposed bootloader analysis tool and the type of IoT device being analyzed. The specific vulnerabilities that this tool can detect will depend on the firmware image being analyzed, but this detects the following IoT vulnerabilities such as:

- Memory corruption vulnerabilities: These include buffer overflow, stack overflow, and heap overflow vulnerabilities.
- Authentication and authorization vulnerabilities include hardcoded credentials, weak or easily guessable passwords, and lack of proper access controls.
- Insecure communication: This includes the use of unencrypted communication protocols or easily guessable encryption keys.
- Insecure data storage: This includes storing sensitive data in unencrypted or easily accessible locations.
- Insecure update mechanisms: This includes unauthenticated or unencrypted update mechanisms that can allow an attacker to install malicious firmware on the device.
- Insecure configuration includes insecure default configurations, such as open network ports or easily guessable default credentials.

- Insecure cryptographic storage: this includes weak encryption algorithms or weak keys.
- Insecure randomness: this includes the use of weak random number generators, which can make encryption keys or session tokens predictable.

The exact steps and the order of the steps depend on the specific implementation of the proposed bootloader analysis tool and the type of IoT device being analyzed. High-level examples and the actual implementation of the algorithm would be more complex, with many more lines of code, more functions, and more parameters. Pseudo Code 1 presents the pseudocode, the actual implementation for the bootloader analysis, and the various types of IoT devices that are analyzed in this research.

Pseudo code 1: analyze IoT firmware

Firmware update pseudo-code in a bootloader for an IoT device is presented in Pseudo Code 2. This pseudo-code is an example of how the firmware update process might work in a bootloader. The bootloader is in an infinite loop waiting for a firmware update command to be received. Once the command is received, it erases the old firmware from memory, receives the new firmware over UART, and verifies it. If the firmware is verified, the bootloader jumps to the new firmware. If the firmware is not verified, an error message is sent over UART, and the bootloader stays in the bootloader waiting for another firmware update command. It's important to note that this is a simplified example, the process will be more complex, and the bootloader will check for any other errors that might happen during the update process and handle them accordingly. Also, this example uses UART for firmware updates, but it can be done over other communication interfaces like TCP/IP, BLE, Zigbee, *etc*.

Pseudo Code 2: Firmware update

Security assessment code for the bootloader for an IoT device is presented in pseudo-code below in Pseudo Code 3. This pseudo-code is a basic example of how the security assessment process might work in a bootloader. The bootloader performs a series of tests to check the memory overflow, firmware integrity, firmware update authenticity, secure boot, and code signing. If any of these tests fail, the bootloader sends an error message over UART and handles the error. It is important to note that this is a simplified example, the process will be more complex, and the bootloader will check for any other security issues that might happen during the assessment process and handle them accordingly.

Pseudo Code 3: Security assessment

The hardware initialization code in a bootloader for an IoT device is presented as Pseudo Code 4. This code initializes the device's memory, communication interfaces, and peripherals, then performs basic tests to ensure the hardware functions properly. If all tests pass, it will jump to the main firmware.

Pseudo Code 4: Hardware initialization code in a bootloader

The firmware file's taint is determined by calculating the taint analysis using the frequency of each byte value in the file. Higher taint levels indicate more random data. This value is a measure of the randomness of the data in the file. Our suggested method computes these to show the firmware image's taint in each area. A bar for each byte value in the Taint represents it in a histogram. The taint value is derived from the distribution of

these frequencies, where the height of each bar indicates the frequency of that byte value in the segment. This program detects the regions of the firmware that might contain compressed or encrypted data by examining the taint of the firmware image. It also can detect any hidden data or malware that might be present in the firmware. The firmware file's taint is determined by adding up the negative odds of every distinct byte value within the file. The number of times a given value occurs divided by the total number of bytes in the file yields the probability of that value.

The taint value is then calculated as presented in Eq. (1):

$$H = \sum (p\_i \times \log(p\_i)) \tag{1}$$

where the probability of the ith byte value is denoted by p_i. All things considered, the authors offer a potent methodology for examining the security of IoT firmware and locating holes in IoT devices' boot processes utilizing a variety of techniques, including Taint analysis, which is covered in the following section.

## RESULTS

The authors performed security assessments which are presented in this section. The hardware setup involved Quad-Core Intel with 64 GB RAM, and 500 GB SD disk running Linux OS for determining taint. The first step involved discovering potentially vulnerable paths in the firmware code which may lead to memory corruption issues. The authors executed assessment scans on different IoT firmware binaries, the initial focus is to determine security vulnerabilities and their sub-classes such as non-volatile memory under the control of the threat vector. The taint calculation is performed on each section of the firmware image, and the taint is displayed as a histogram. The x-axis of the histogram represents the byte values, and the y-axis represents the frequency of each byte value in the section. The taint value is calculated from the distribution of the byte values and is displayed as a single value for each section of the firmware image. By analyzing the taint of the firmware image, the tool identified areas of the firmware that may contain encrypted or compressed data, as well as potentially identifying any hidden data or malware that may be present in the firmware. This is because encrypted or compressed data will have a higher tint value, due to the randomness of the data. On the other hand, data that is not encrypted or compressed will have a lower taint value and may indicate the presence of structured data, such as executable code or file systems.

The second step is based on a test-execution environment for detecting unique vulnerabilities using taint analysis. This can be executed using an HTTP response containing variables. The analysis's entry points, or sources, are places in the program where unreliable, user-controlled data can enter the code being examined, such as when reading from the standard input or environmental variables. The analysis's conclusion points are referred to as sinks, and they represent security-sensitive actions that could be used by attackers to launch attacks, such as jump instructions for obstructing intended control flow. Data that is unreliable is flagged during analysis by becoming tainted, and the taint is subsequently spread across the code by a taint propagation policy. If a sink runs operations on contaminated data, vulnerabilities are found, and app integrity is

**Code & Taint Expressions**

$Seed\_function(taint\_y)$

$x = taint\_y$

$taint\_y = Taint\_taint\_y$

$x = deref\ (TAINT\_taint\_y\_loc\_x)$

**Taint Memory**

taint_y
x

**Tainted Page**

**Figure 3 Taint propagation.**               

compromised before being discovered. The response template is generated using a fuzzer using probabilistic-context-free grammar as a tuple denoted in Eq. (2) as

$$A = (Nt,\ St,\ Pt,\ St) \tag{2}$$

where Nt = Set of non-terminal symbols

St = Set of terminal symbols

Pt = Production rules

St = Starting symbol

The taint propagation is illustrated in symbolic form in Fig. 3 below.

After applying this approach to five different IoT devices running at least two different applications, the authors selected the below applications to check for communication challenges. Three of these are Firebase and one each with NFC, Bluetooth, and Internet. In four out of five cases, the authors discovered potentially dangerous flaws. For example, the doorbell camera presents the dataflow of the picture of a person at the door to the owner's camera or mobile over the Internet. Table 2 presents the different vendor firmware that are assessed in this research.

Similar flows were discovered in other cases having potentially dangerous flows, yet those were needed to implement the main functionality of the device application. Only the Auto Assistant was secure, made multiple checks of the values, and sanitized all elements involved in the dataflow as presented in Table 3.

Vulnerability warnings are illustrated in Fig. 4 issued warnings about potential malicious injections, these correspond to threat and privacy issues, the first warning relates to sensitive data with the injection method 'execute' creating an HTTP request. The taint analysis detects the next warnings leading to geolocation triangulation and finally concatenates and points to a web service as URL, this points to the app programs and bandwidth and location of the IoT device. This is potentially a huge privacy breach.

Table 4 presents a summary of multiple malwares captures and malicious scenarios obtained after executing Zeek-based network analysis on the IoT ecosystem comprising of the smart doorbell, electric monitor, color thing, BLE energy, and auto-assistant devices.

In our research, we utilized Zeek, known as Bro, an open-source network security monitoring tool, to conduct a comprehensive analysis of network traffic generated by various Internet of Things (IoT) devices. The focus of our study involved understanding the dynamics and interactions within an IoT ecosystem through the lens of network traffic analysis. The IoT environment consisted of diverse device categories, each contributing unique functionalities to the network. The ecosystem encompassed several types of IoT

**Table 2 Vendor device information.**

| Type | Firmware version | Product | Last modified |
|---|---|---|---|
| Wifi doorbell | 1.07 | Qubo smart Wifi | 21 April 2018 |
| Electric monitor | v101-r018 | Amici sense | 21 Dec 2020 |
| Color thing | FW1.07B09 | NFC | 10 Sept 2021 |
| BLE energy | AR401X_REV6 | Bluetooth | 22 Dec 2017 |
| Auto assistant | BACnet_4.2 | Bluetooth | 15 July 2020 |

**Table 3 Device dataflow information.**

| Device app | Channel | Edge | Mobile | Dataflow |
|---|---|---|---|---|
| Smart doorbell | Firebase | 13.67" | 55.34" | Surveillance cam to user mobile app |
| Electric monitor | Firebase | 115.23" | 54.67" | From timestamp to user's mobile app |
| Color thing | NFC | 13.19" | 78.67" | From input by user to LEDs |
| BLE energy | Bluetooth | 15.13" | 87.45" | From shared preference to user mobile |
| Auto assistant | Internet | 45.57" | 51.23" | No issue found |

```
#1 CheckWifiTask.java:110:XSS-injection method "execute"
#2 CheckWifiTask.java:113:Log forge method "w"
#3 CheckWifiTask.java:116:Log forge method "d"
#4 FetchingAlertsTak.java:168:Log forge method "d"
#5 FetchingAlertsTak.java:176:URL injection method "initialize"
```

**Figure 4 Taint vulnerability warning.**

**Table 4 Summary of the benign scenarios.**

| Dataset | Attack duration (in hours) | Packets (in thousand) | Zeek flows | PACP size (in MB) | Device |
|---|---|---|---|---|---|
| IoT_Cap-1 | 12.7 | 9.276 | 138 | 2.965 | Smart doorbell |
| IoT_Cap-2 | 15.8 | 14.298 | 245 | 4.761 | Electric monitor |
| IoT_Cap-3 | 20.1 | 11.567 | 589 | 8.242 | Color thing |
| IoT_Cap-4 | 19.5 | 20.452 | 421 | 3.789 | BLE energy |
| IoT_Cap-5 | 22.5 | 8.451 | 789 | 5.783 | Auto assistant |

devices, ranging from smart doorbells to electric monitors, color-changing devices (referred to as 'color things'), Bluetooth Low Energy (BLE) devices for energy monitoring, and voice-activated assistants. Smart doorbells, equipped with cameras and internet connectivity, enable remote door monitoring and interaction. Electric monitors are systems that observe and transmit electricity consumption data for analysis and management. Throughout our network analysis using Zeek, we encountered instances of malware captures, signifying the presence or attempted infiltration of malicious software within the network. These captures denote records or logs of suspicious or potentially harmful network activities observed during the analysis. Additionally, our research

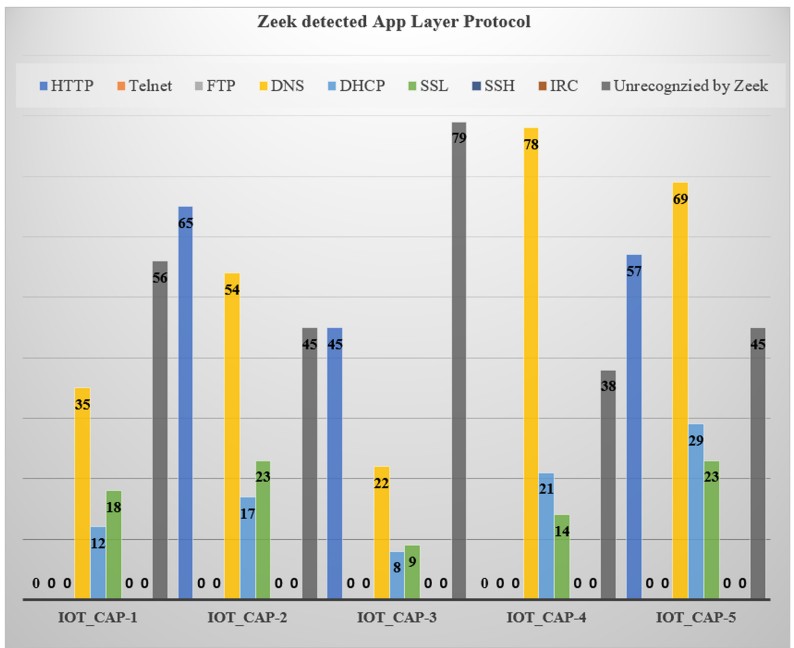

**Figure 5 Application layer protocol breakdown summary.**

highlighted specific malicious scenarios where vulnerabilities within the IoT ecosystem were exposed or when instances of malware attempted to compromise the devices in our study. These scenarios were documented to showcase potential risks and vulnerabilities prevalent in the IoT landscape, elucidating the need for robust security measures within these networks.

The malware attacks were executed over a long period, rotated every 24 h, and the network traffic was captured in the form of a PCAP file. However, in a few cases, the captured traffic PCAP grew very fast, and the captures were stopped before 24 h, so some of the PCAPs differ in the capture durations as presented in Fig. 5.

For advanced-level analysis, the authors scanned and enumerated each infected device at application layer prediction by filtering and summarizing the Zeek information. In this, the number of dataflows as per the protocols (such as HTTP, DNS, DHCP, Telnet, SSL, and IRC were filtered) and some were not recognized where the flow was not quantifiable, as presented in Fig. 6.

From these taking and malware attack scenarios, the benign IoT network datasets were obtained, including information regarding the duration of the attack, packets involved, Zeek flow, PCAP file, and device, as presented in Table 5 below.

The compromised datasets using the attack methodology were validated for taint analysis. This research taint analysis of IoT devices uncovered several vulnerabilities as presented in Table 6.

These are just a few samples from the research performed for the results that are obtained from taint analysis of IoT device firmware using datasets from Stratosphere IPS.

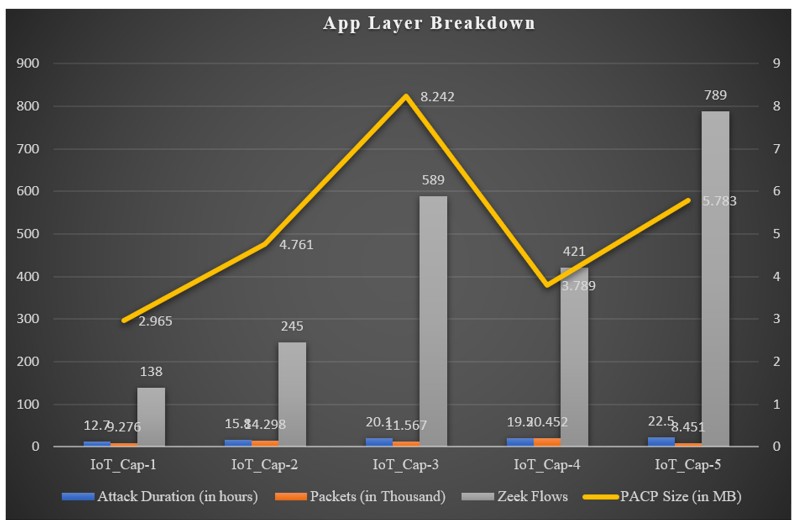

**Figure 6 Application layer breakdown for benign attacks.**

**Table 5 Application layer protocol breakdown for benign scenarios.**

| Dataset | HTTP | Telnet | FTP | DNS | DHCP | SSL | SSH | IRC | Unrecognized by Zeek |
|---------|------|--------|-----|-----|------|-----|-----|-----|----------------------|
| IoT_Cap-1 | – | – | – | 35 | 12 | 18 | – | – | 56 |
| IoT_Cap-2 | 65 | – | – | 54 | 17 | 23 | – | – | 45 |
| IoT_Cap-3 | 45 | – | – | 22 | 8 | 9 | – | – | 79 |
| IoT_Cap-4 | – | – | – | 78 | 21 | 14 | – | – | 38 |
| IoT_Cap-5 | 157 | – | – | 69 | 29 | 23 | – | – | 45 |

**Table 6 Datasets validated for taint analysis.**

| Dataset | AD | BO | IV | DL | ID | UA | Device |
|---------|-----|-----|-----|-----|-----|-----|--------|
| IoT_Cap-1 | 12.7 | C | N | C | C | C | Smart doorbell |
| IoT_Cap-2 | 15.8 | C | C | N | N | C | Electric monitor |
| IoT_Cap-3 | 20.1 | C | C | C | N | C | Color thing |
| IoT_Cap-4 | 19.5 | N | C | C | C | C | BLE energy |
| IoT_Cap-5 | 22.5 | C | C | N | C | C | Auto assistant |

**Note:**
AD, Attack duration in hours; BO, buffer overflow; IV, input validation; DL, data leakage; ID, information disclosure; C, compromised; N, not compromised.

# CONCLUSIONS

One essential component of IoT security is IoT device taint analysis, which helps to detect security flaws in IoT devices and thwart malicious attacks. To conduct taint analysis, researchers can find a variety of vulnerabilities in IoT devices, such as insecure input validation, buffer overflows, data leakage, unauthorized access, and information disclosure by utilizing datasets. The design and execution of bootloaders present major hurdles, despite their crucial role in the security and functionality of Internet of Things devices.

Additional investigation is required to tackle these issues and create more effective and safer bootloading systems for smart IoT devices. All things considered, bootloaders are essential to the operation and security of embedded and Internet of Things devices. They oversee setting up security mechanisms to prevent unwanted access or manipulation, initializing the device's hardware, and loading the primary operating system or firmware.

## DISCUSSION AND FUTURE SCOPE

The continued growth and complexity of IoT devices will require ongoing research and development of taint analysis techniques to stay ahead of potential threats. IoT device taint analysis is a vital tool for ensuring the security and reliability of IoT devices and will continue to play a critical role in protecting against malicious attacks in the future.

### Funding

This work was supported by the Deanship of Scientific Research at King Khalid University through the large Groups Research Project under grant number (RGP.2/175/44). The funders had no role in study design, data collection and analysis, decision to publish, or preparation of the manuscript.

### Grant Disclosures

The following grant information was disclosed by the authors:
King Khalid University: RGP.2/175/44.

### Competing Interests

The authors declare that they have no competing interests.

### Author Contributions

- Akashdeep Bhardwaj conceived and designed the experiments, performed the experiments, performed the computation work, prepared figures and/or tables, and approved the final draft.
- Ankit Vishnoi conceived and designed the experiments, performed the experiments, analyzed the data, performed the computation work, prepared figures and/or tables, and approved the final draft.
- Salil Bharany conceived and designed the experiments, performed the experiments, analyzed the data, performed the computation work, prepared figures and/or tables, authored or reviewed drafts of the article, and approved the final draft.
- Abdelzahir Abdelmaboud performed the computation work, prepared figures and/or tables, authored or reviewed drafts of the article, and approved the final draft.
- Ashraf Osman Ibrahim performed the experiments, analyzed the data, prepared figures and/or tables, authored or reviewed drafts of the article, and approved the final draft.
- Mohamed Mamoun analyzed the data, prepared figures and/or tables, authored or reviewed drafts of the article, and approved the final draft.

- Wamda Nagmeldin analyzed the data, prepared figures and/or tables, authored or reviewed drafts of the article, and approved the final draft.

## Data Availability

The code is available in the Supplemental Files.

## Supplemental Information

Supplemental information for this article can be found online at http://dx.doi.org/10.7717/peerj-cs.1771#supplemental-information.

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
