# Peer review of "Framework to perform taint analysis and security assessment of IoT devices in smart cities"

_PeerJ Computer Science, doi:10.7717/peerj-cs.1771_

## Round 0.1 · original submission · Major Revisions

The authors should prepare a major revision and rebuttal to reply to all comments sent by the reviewers.

·

Basic reporting

The article, "Framework to Perform Taint Analysis and Security Assessment of IoT Devices in Smart Cities" is a novel, clear, and very good idea to be published in PeerJ Computer Science but it requires some changes. For example, some recent review papers have not been included for the Security of the Internet of Things, please add them in your references to make the paper self-sufficient:

Recommended References:
1. https://ieeexplore.ieee.org/stamp/stamp.jsp?tp=&arnumber=8742551
2. https://www.sciencedirect.com/science/article/pii/S2214212617302934
3. https://www.mdpi.com/1999-5903/12/3/55
4. https://ieeexplore.ieee.org/abstract/document/9902998
5. https://ieeexplore.ieee.org/abstract/document/8666588

Experimental design

The proposed schema is good.

Validity of the findings

Please make some comparative experiments to further validate the superiority of the scheme with some other works in the literature to bolster the experimental findings.

Additional comments

Please add some running time experiments, show power consumption details for the simulator, and also show the elapsed computational time as it is an important parameter for the resource-constrained IoT.

·

Basic reporting

The paper presents an interesting area of research. The authors have tried to showcase the taint analysis of IoT devices by presenting an algorithm and justifies by showing the results. Following are my comments/suggestions:

In the introduction, the authors have claimed the statements about bootloader, firmware, and it's component withtout using a single citation. The identified research gap does not come from the limitations introduced in the introduction. There is a need of improving the flow of the introduction and connection of the research gap with some existing limitations.

The related work, the short forms (FSDS, COTS, and others) are used directly without any prior knowledge. The critical analysis of the existing work is significantly missing. There is a need of huge improvement in the related work. For example, a summarized table can be added with pros and cons of the existing work while comparing it with the proposed methodology. More recent papers needs to be added that are directly related to the firmware security or the attacks that are targeted in the research.

The quality of figures needs to be improved. I do not understand why the authors have presented algorithms in the form of table? Is it the guideline of journal submission?

The explanation of results does not conclude the statements provided in the introduction. The authors need to map the research gaps with the results provided in the paper.

Experimental design

The authors did not mention any research questions.
The methods and materials used in the paper is ambiguous. I do not understand if the authors wants to target all the discussed vulnerabilities or these are the possible ones? In the latter case, there is not a single citation. Moreover, which tool is responsible to detect these vulnerabilities? The details are not mentioned. If it is the one that the authors have proposed then, it needs to be clear in the text.
The authors have mentioned such tools are hugely valuable then, they explained the steps to solve the problem. It creates a huge ambiguity in understanding the contribution of the authors.
In the methodology, the authors should mention which vulnerabilities are they targeting and explain how their proposed algorithm solves the problem.
The authors should include a figure of their overall framework which shows the complete work flow of their methodology. The presented figures are quite general.
I don't know from nowhere the zeek detected app layer protocol. The explanation of results needs to be clear.

Validity of the findings

If the methodology and results are focused and mapped with the research questions then the findings will be valuable. There is no doubt the authors have targeted an extremely important problem in the field of IoT security. I

Additional comments

There is no doubt the authors have targeted an extremely important problem in the field of IoT security. I encourage the authors to apply the changes to improve the readability and quality of paper in terms of technical and grammatical mistakes as well.

---

## Round 0.2 · Major Revisions

The reviewers indicate that the article requires major revisions as comments from the first round were not handled completely.

**Language Note:** The review process has identified that the English language must be improved. PeerJ can provide language editing services - please contact us at copyediting@peerj.com for pricing (be sure to provide your manuscript number and title). Alternatively, you should make your own arrangements to improve the language quality and provide details in your response letter. – PeerJ Staff

·

Basic reporting

No comments

Experimental design

No comments

Validity of the findings

No comments

Additional comments

No comments

·

Basic reporting

The comments provided in the earlier review cycle are not incorporated properly.
- The references to support the arguments in the introduction are not incorporated.
- The table of summarized related work is not included.
- The related work does not support directly to the research gaps.
- The results are not included.

Experimental design

The experimental set up is defined theoretically however, there is a need of in depth analysis. The algorithms provided in the materials section are only functions. The definitions are not included.

Validity of the findings

Even it was mentioned in the comments before that the research gap should be mapped with the results and methods. The paper still lacks the above-mentioned point.

Additional comments

There are many grammatical mistakes. The flow is missing from related work to the end of the paper.
The paper has a potential as it covers state of the art topic. It needs a lot of improvement.

---

## Round 0.3 · accepted · Accept

The article is accepted after 2 rounds of revision while addressing reviewer comments.

·

Basic reporting

The overall understanding of the paper has improved. However, there are many grammatical mistakes.

Experimental design

The methodology looks ok. The authors have incorporated the suggested changes.
The steps by step experimental approach has added value for those who wants to reproduce the paper.

Validity of the findings

Satisfied.